# The Cross-Border Mergers and Acquisitions of Local State-Owned Enterprises: The Role of Home Country Government Involvement

**Qiuyang Gu** [1,2,3], **Chunhua Ju** [1,2,3] **and Fuguang Bao** [1,2,3,*]

1   Department of Modern Business Research Center, Zhejiang Gongshang University, Hangzhou 310018, China;
    guqiuyang123@163.com (Q.G.); jch@zjgsu.edu.cn (C.J.)
2   School of Management Science & Engineering, Zhejiang Gongshang University, Hangzhou 310018, China
3   School of Business Administration, Zhejiang Gongshang University, Hangzhou 310018, China
*   Correspondence: baofuguang@126.com or bfg@zjgsu.edu.cn

**Abstract:** Existing literature tends to treat enterprises as a whole when measuring government intervention. However, in Chinese region-specific institutional development, ultimate control (i.e., local government) tends to control multiple enterprises. This paper considers the enterprises controlled by the same ultimate controller as a portfolio, which is used to measure government intervention by comparing the differences of the enterprises in the portfolio. This paper uses the data of Chinese listed local state-owned enterprises (LSOEs). and we assess whether local state ownership benefits or offsets LSOEs' cross-border mergers and acquisitions (CBM & A) activities. We propose a new measurement of government intervention to explain the mechanisms through which government influences the cross-border mergers and acquisitions of local SOEs. The experimental results show that government intervention and region-specific marketization institutional development negatively moderate the effect of government internationalization subsidies and government intervention on CBM & A separately. However, government internationalization subsidies, government intervention, and region-specific marketization enhance the CBM & A effect of state ownership separately. This study explores the benefits of government involvement in local SOEs. The value of this paper is to provide a novel perspective, including the intermediary effect of government intervention and the market environment.

**Keywords:** cross-border mergers and acquisitions; government involvement; home country government involvement; state-owned enterprises

## 1. Introduction

Due to the increasing competitiveness of Chinese local state-owned enterprises (SOEs), more and more local SOEs go abroad to achieve rapid growth through cross-border mergers and acquisitions (CBM & A) [1,2]. According to the Chinese bulletin of the FDI (Foreign Direct Investment) in 2016, after 2012, Chinese SOEs' CBM & A were on the fast track, with the average annual SOE's CBM & A kept at more than $40 billion. By 2017, 765 Chinese SOEs M&A projects had been implemented, involving 74 countries (regions), with the actual transaction volume $35.33 billion. CBM & A projects involve manufacturing, software and information technology, transportation, and another 18 industries. In the context of the continuous expansion of Chinese enterprises' CBM & A, the economic effect of Chinese SOEs' CBM & A has become an increasingly concerning issue [3].

Within the current historical background of further deepening the reform of central and local SOEs and the comprehensive upgrading of industrial structure, M&A have become some of the most important means of "strengthening, optimizing, and enlarging" state-owned capital [4]. According to

Wind statistics, in 2017 alone, 126 state-owned listed companies underwent mergers and acquisitions, with transactions worth 853.8 billion yuan. With the constant development of Chinese SOEs, more and more central and local SOEs go abroad to achieve rapid growth through CBM & A [5]. Because of the background of our country's transition economy and the ownership structure of state-owned shares, the local government can easily be involved in the operation of local SOEs with a dual status shared with the owner of local state-owned assets and social managers. Compared with the central government, the possible reasons for local governments to be involved in local SOEs' CBM & A include the following: First, local governments acquire overseas high-tech enterprises through state-owned listed companies and introduce advanced technologies [6]. Second, under the promotion mechanism of financial assessment based on relative performance evaluation, government officials during their tenure tend to merge local SOEs with overseas enterprises to prevent resource loss and promote performance improvement [7].

Academic circles have always believed that government intervention is an important behavior that affects enterprises [8]. The existing literature contains indicators to measure the degree of government intervention, mainly including property rights, the proportion of state-owned shares, the district institutional environment, the political affiliation of an executive or director, pyramid hierarchy, etc. [9–11]. Although CBM & A are very important for Chinese local SOEs, research of the effect of CBM & A on local SOEs is still very rare. Most of the relevant studies only consider state-owned equity as a factor, and the research subject is not state-owned enterprises themselves. SOEs' inherent characteristics of bureaucracy and low efficiency seem to make the CBM & A of SOEs doomed, and it is difficult to achieve a good effect, but Reddy, et al. [12] found with China and India in the nearly 20 years from 1995 to 2014 a higher value when tracking studies of CBM & A. Compared with local private enterprises, Chinese local SOEs which have completed CBM & A have a higher value in the trading process. To study the motivation of SOEs' CBM & A, a key question needs to be answered: what drives value creation in SOEs' CBM & A? There is no answer to this question from the existing research.

Our analysis helps determine whether government involvement affects the CBM & A activities of state-owned enterprises. First, while prior research realized that strong relationships with governments help SOEs do CBM & A, little research has tested the influence of local-level government involvement on local SOEs' CBM & A [13]. Second, although the literature often assumes that government actions will affect the CBM & A of state-owned enterprises, little literature mentions that such an influence will be affected by the degree of marketization. Third, the existing literature only examines the influence of government behaviors on the CBM & A of SOEs. They do not discuss whether the CBM & A of different types of SOEs (central-level or local-level government holding) are all influenced by the government.

The structure of this paper is as follows: Section 2 reviews the related literature and puts forward the hypothesis. Section 3 describes the data source, variable construction, and descriptive statistics. Section 4 describes the empirical results. Section 5 includes additional tests. Section 6 shows the conclusion of this paper and future work.

## 2. Literature Review and Hypothesis Development

According to traditional agency theory, state ownership has a limited effect on stimulating firms' capabilities. In reality, however, several SOEs in China have changed to "energetic", rather than "dying dinosaurs" [6,14,15]. There are more than 115 Chinese companies in the 2017 Fortune Global 500, about three-fifths of which are SOEs. Different from privately owned enterprises (POEs), SOEs tend to be risky [16], have more informal performance evaluation and compensation systems for top managers [17], and have priority in resource allocation [18]. Recently, SOEs have also played an important role in globalization through foreign investment and exports. Changes in the national economic environment and regional environment have an important influence on the CBM & A activities of local SOEs [3]. A recent study shows that Chinese SOEs are suffering more complex pressures than private enterprises in host countries. The reason is likely to be due to ideological conflicts and the potential threat of a price war, etc. [19]. A study shows the model of steering and

monitoring SOEs, which can group SOEs by the number of state shareholder owners and the proportion of state ownership [20]. In more than 60% of the CBM & A activities of SOEs, the maximization of shareholder value is the core element and strength. The above phenomenon is very similar to the purpose of the CBM & A of private enterprises. However, another 40% of the CBM & A activities of SOEs mainly aim at the following three points: financial distress, capital devouring, and shareholder benefit maximization, among which financial distress is the most important and shareholder benefit maximization is the least important [9]. From the financial perspective of Chinese SOEs, the basic purposes of CBM & A activities can be divided into the following two varieties: the optimal interests of shareholders and the maximization of interests of other stakeholders (corporate managers, corporate creditors, and business-related parties). In the first case, SOEs improve the profitability of the company through M & A activities and finally achieve the purpose of increasing the shareholder value of the company. In the second case mentioned above, SOEs actually carry out M&A activities with a certain stakeholder as the leading party [21].

Government internationalization subsidies have a strong effect on SOEs carrying out external expansion [22]. The existing studies used different proxies to measure the influence of government involvement on firms' CBM & A, e.g., a SOE dummy variable [23], state-owned capital over total capital [7], region-specific home institutional development [24], political connections [25], and the pyramid structure of SOEs [26]. However, these studies tend to regard all the firms as an individual agent, while in China, the government (ultimate controller) usually controls multiple SOEs at the same time. Studies show that the performance of SOEs in developed countries is not as good as that of non-state-owned enterprises in terms of stock price or business performance [27]. Busse and Hefeker [28] combined agency theory with institutional analysis. They proposed a mechanism for state-owned enterprises to expand abroad by introducing state subsidies. Alon et al. [29] showed that Chinese SOEs are likely to significantly increase overseas investment in the short to medium term. Another study showed that state-owned enterprises are decisive in making decisions about whether to enter international markets. This is because if the purpose of the international expansion of state-owned enterprises is to adjust their strategies to adapt to the changing system and market environment, the international expansion of SOEs is consistent with the goal. Although institutional pressure or institutional support will have different impacts on the CBM & A process of all enterprises in the economy [13], the influence of SOEs and private enterprises varies in different aspects. The reason is the different institutional compatibility they face in their home country and abroad.

**H1.** *Government internationalization subsidies have a positive association with firms' CBM & A activities.*

Considering other factors of government intervention, Xia and Chen [16] pointed out that the decentralization at central and local levels in China, as well as the decentralization between state-owned and local governments and their affiliated SOEs, changed the interaction mode and connection between the government and SOEs. On the one hand, governments at all levels control important SOEs in the form of direct holdings or a high proportion of ownership shares. This kind of phenomenon manifests in important industries such as the military industry and the petrochemical industry. On the other hand, governments at all levels have loosened their control over small or uncompetitive SOEs by indirectly controlling or holding a lower proportion of shares. There is a high degree of marketization of this kind of phenomenon in the Chinese eastern region more generally; by reducing government intervention in the jurisdiction of local government, there is active relaxation of the intervention and control of SOEs, so as to reduce the local SOEs' performance evaluation of invalid information interference and improve executive efficiency, in order to reduce the problem of the agent of local government intervention. In eastern China, where the non-state-owned economy is also active, private and foreign-funded enterprises can also report rent-seeking behaviors of SOEs to local government authorities to ensure the effectiveness of fair competition [30,31]. In China, since the implementation of administrative decentralization in the early 1980s and early 90s, fiscal decentralization reform, fiscal autonomy, and the administration of local government have strengthened increasingly, gradually

forming federalist "Chinese characteristics" [32]. This, on the one hand, makes the enthusiasm of local government have greater effect on development in the regional economy, but on the other hand, it also makes the local government have more power within a local protection policy.

In view of the above literature, this paper analyzes how state control affects the degree of SOEs' globalization from the perspective of SOEs' globalization. We intend to study this issue from the perspective of the national governance mechanism of the relationship between managers of SOEs and their global decision making. As the largest emerging economy in the world, managers of local SOEs are often selected and appointed by the state directly from local government officials [33]. These SOEs' executives and their SOEs carry out foreign business expansion and M&A under the guidance and capital control of local governments [31]. However, the motivation for senior managers of SOEs to promote a SOE's CBM & A is not only to improve the economic performance of local governments but also to achieve political goals and social welfare goals at the national level [34]. Li et al. [35] show that the incompatibility between China's political system and foreign political market system reduces the difficulty of Chinese SOEs in outbound investment and M&A activities. Secondary political factors in China, such as coercive measures, standardized management, and the imitation effect, offset the advantages brought by M&A. Du and Boateng [36] adopt the incident research method to study the CBM & A behavior of Chinese SOEs. For M&A players in China, the government and system have a significant influence on the improvement of financial performance after CBM & A. The impact of government intervention on M&A performance can be further divided into two aspects. First, direct government intervention mainly refers to the role of government policies [37]. For instance, during the financial crisis, the ministry of commerce issued a series of supportive policies, including providing resource support, low-interest loans, tax breaks, and other fiscal policies to create conditions for enterprises to go abroad. At the same time, the complex application process increased the cost of mergers and acquisitions. Direct government intervention is closely related to M&A costs [38]. Second, the government's indirect intervention is reflected in the government's intervention in enterprises' M&A through SOEs [35].

Studies show that state-owned enterprises in different regions have different execution efficiency and evaluation mechanisms when they are subject to different levels of government intervention. State-owned enterprises that have received less government intervention, that is, state-owned enterprises under indirect government control with a low shareholding ratio and local state-owned enterprises from non-regulated industries or the eastern seaboard, are more inclined to use a market-oriented performance evaluation method, that is, paying more attention to the performance evaluation of the senior executives of SOEs. Executives of SOEs who face strong interference from the government are more inclined to use government-oriented evaluation methods for performance evaluation [39]. In addition, the marketization level restrains or even reverses the intervention effect of local government on local SOEs' M&A. However, when the government has stronger control over local SOEs, the marketization level increases the inhibition of local SOEs' M&A, especially on share-diversified M&A [5]. The government is more willing to encourage the M&A activities of SOEs to promote SOEs' investment in certain fixed assets and major national infrastructure projects as well as the control of important strategic resources, including oil, gas, and minerals. However, the government's intervention in local state-owned investment is mainly reflected in local SOEs, while the performance of central SOEs or private enterprises is not obvious [1]. Zhao, Chen, and Hao [11] believe that China's monetary policy and local regulations have different impacts on SOEs and non-state-owned enterprises. Specifically, local government intervention can weaken, reverse, or even distort the effect of monetary policy and significantly alleviate the administrative purpose and investment expectation of local SOEs, which are largely controlled by local governments.

**H2a.** *Government intervention has a positive relationship with firms' CBM & A.*

**H2b.** *Government intervention has a positive mediation effect on the relationship between government internationalization subsidies and firms' CBM & A.*

Lehmann and Benner [40] describe the theory and mechanism of the impact of region-specific institutional development on corporate behavior. Coluccia's [41] experimental conclusion shows that the external expansion behavior of SOEs is influenced by different institutional factors and external factors, such as supervision intensity, accounting system norms, external cultural influence, and the attitude of corporate stakeholders. These forces do have a systematic and coercive impact on a company's external expansion. In Cai and Sevilir's [42] study, region-specific institutional development has a positive relationship with the entrepreneurial success rate. They also pointed out that, due to differences in geographical location, resource characteristics, and related policies, economic development paths in different regions in China are different and the degree of marketization is unbalanced. In different marketization environments, there are great differences in the business decisions made by enterprises. As the degree of the external marketization of SOEs continues to improve, the market environment and conditions of M&A activities have been improved. When state-owned enterprise ownership concentration is low, the external market environment makes the other big shareholders able to effectively supervise government behavior, to a certain extent inhibiting the political or social objectives of the impact of an enterprise's M&A decision, prompting M&A decision-making to be more suitable for enterprise development and economic development, with mergers and acquisitions performance able to be improved as a whole [33,43].

Existing studies also show that enterprises with weak market institutional backgrounds have more incentive to expand and acquire overseas to seek a wider market and higher profits. Obviously, the development level of SOEs within the same industry varies in different regions and political environments [44]. To capture this effect, we followed Wang et al. [4] and Wu et al. [24] to include region-specific home institutional development. This operationalization adopts the marketization index developed by Li and Ding [45] believed that the lack of a support system or market allocation would bring a competitive disadvantage to Chinese local SOEs. This is embodied in the following five key factors: the level of economic development, degree of market tolerance, basic setting and allocation, degree of development of a factor market, and the development potential of a free market. The higher the degree of marketization, the higher the market maturity of the region.

**H3.** *The better the region-specific institutional development, the weaker the effects of government intervention on a firms' CBM & A.*

With the above background, it might be appropriate to treat the firms controlled by the same ultimate controller as a portfolio and measure the extent of government intervention by comparing firms' proportions within the portfolio. Moreover, we applied the propensity score matching (PSM) and instrumental variables (IVs) to further alleviate the concerns of endogeneity and the causal selection problem. The use of PSM allowed us to minimize institutional and industrial structural differences between local state-owned enterprises and local private enterprises. The number of years since the city was forced to trade and regional GDP per capita in the regressions were used as IVs. The theoretical research model is summarized in Figure 1. Figure 1 shows the brief model for this research. Government internalization subsidies and government intervention in Figure 1 represent the independent variable in this research. Local state-owned enterprises cross-border mergers and acquisitions in Figure 1 represents the dependent variable in this research. Institutional context (region-specific home institutional development) in Figure 1 represents the moderator variable in this research. All variables in the control variable part represent the control variable in this research. All the details of control variables and related literature are detailed in Section 3.

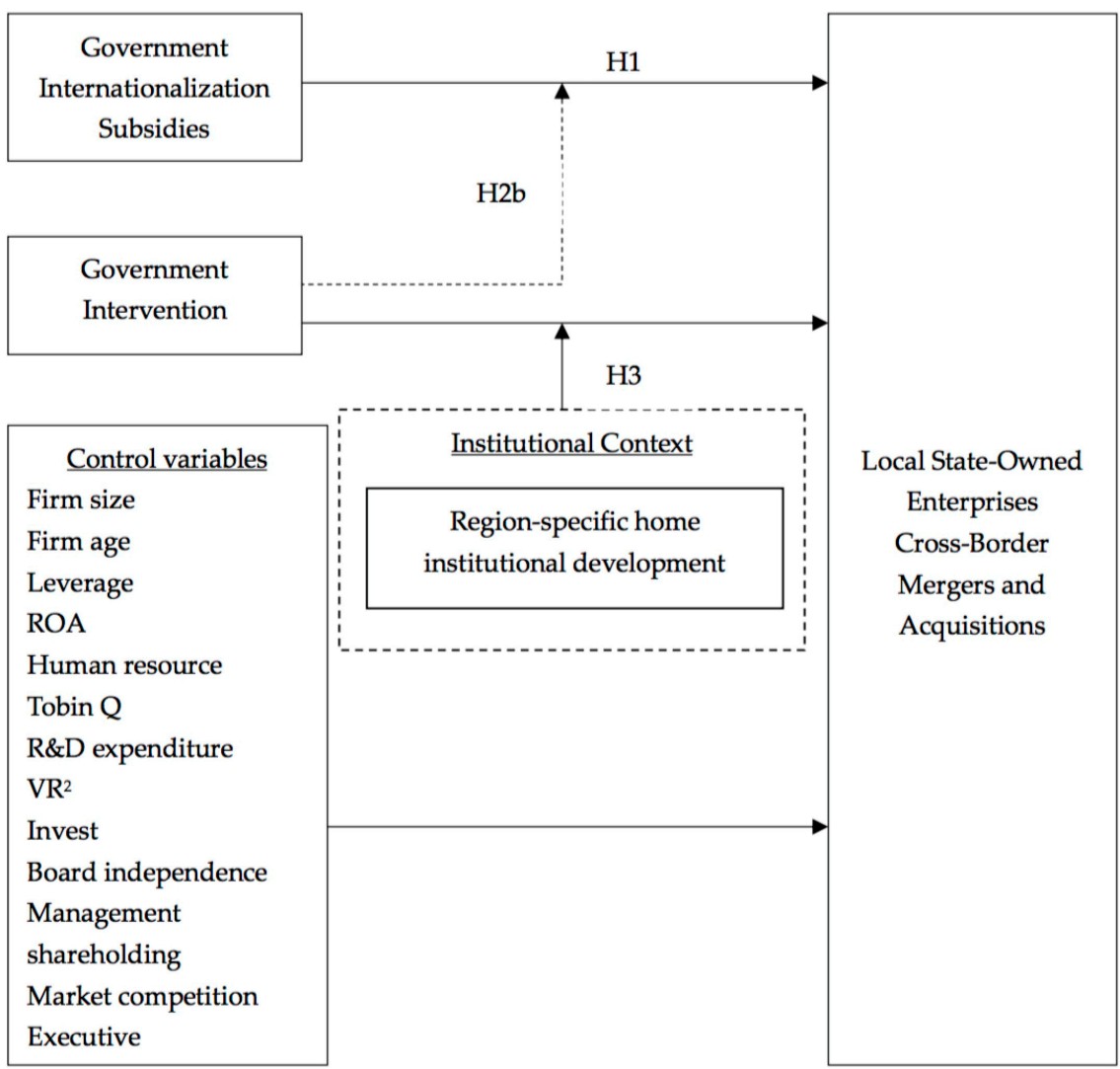

**Figure 1.** Research Model.

## 3. Data Variable Construction and Descriptive Statistics

We constructed a dataset for local state-owned companies trading on the Shanghai and Shenzhen Stock Exchange that have been involved in outward FDI and exporting activities between 1998 and 2017 (see Tables 1 and 2 for details). Information regarding Chinese local SOEs' outward CBM & A activities and firm-specific information, such as firm age, size, number of employees, staff training expense, ownership structure, return on assets (ROA), and R&D expenditure, was obtained from the CSMAR database (CSMAR database is a professional economic and financial data platform.). We extracted firm exporting and executives (name, age, and education background) data from the Wind database, along with information on different types of government subsidies, government intervention, ROA, asset-liability ratio, Tobin's Q, export subsidies, financial subsidies, and other subsidies. What is worth paying attention to is the Tobin's Q selected in this paper was used with the arithmetic mean. We also supplemented it with another Chinese database reset and manually checked and supplemented.

**Table 1.** Sample Composition.

| Year | Number of Announcements | Percentage | Number of Firms | Percentage |
|---|---|---|---|---|
| 2003 | 9 | 2.39% | 9 | 3.16% |
| 2004 | 6 | 1.60% | 5 | 1.75% |
| 2005 | 15 | 3.99% | 10 | 3.51% |
| 2006 | 17 | 4.52% | 16 | 5.61% |
| 2007 | 26 | 6.91% | 23 | 8.07% |
| 2008 | 42 | 11.17% | 34 | 11.93% |
| 2009 | 62 | 16.49% | 41 | 14.39% |
| 2010 | 38 | 10.11% | 33 | 11.58% |
| 2011 | 31 | 8.24% | 24 | 8.42% |
| 2012 | 12 | 3.19% | 6 | 2.11% |
| 2013 | 15 | 3.99% | 13 | 4.56% |
| 2014 | 18 | 4.79% | 14 | 4.91% |
| 2015 | 19 | 5.05% | 12 | 4.21% |
| 2016 | 9 | 2.39% | 6 | 2.11% |
| 2017 | 57 | 15.16% | 39 | 13.68% |
| Total | 343 | 100.00% | 270 | 100.00% |

Notes: This table reports full sample distribution by year. We report the number of announcements and the involved firms. Panels span the period 2003–2017.

**Table 2.** Portfolio Composition.

| Number | Observation | Proportion (%) | Number | Observation | Proportion (%) | Number | Observation | Proportion (%) |
|---|---|---|---|---|---|---|---|---|
| 1 | 1015 | 12.101 | 16 | 144 | 1.717 | 33 | 99 | 1.180 |
| 2 | 660 | 7.868 | 17 | 85 | 1.013 | 35 | 35 | 0.417 |
| 3 | 633 | 7.546 | 18 | 144 | 1.717 | 36 | 36 | 0.429 |
| 4 | 416 | 4.959 | 19 | 266 | 3.171 | 37 | 37 | 0.441 |
| 5 | 320 | 3.815 | 20 | 60 | 0.715 | 58 | 58 | 0.691 |
| 6 | 372 | 4.435 | 21 | 63 | 0.751 | 60 | 60 | 0.715 |
| 7 | 616 | 7.344 | 22 | 110 | 1.311 | 61 | 61 | 0.727 |
| 8 | 280 | 3.338 | 23 | 138 | 1.645 | 62 | 124 | 1.478 |
| 9 | 261 | 3.112 | 24 | 24 | 0.286 | 63 | 63 | 0.751 |
| 10 | 230 | 2.742 | 25 | 25 | 0.298 | 64 | 128 | 1.526 |
| 11 | 220 | 2.623 | 26 | 52 | 0.620 | 65 | 195 | 2.325 |
| 12 | 360 | 4.292 | 27 | 27 | 0.322 | 67 | 67 | 0.799 |
| 13 | 312 | 3.720 | 29 | 29 | 0.346 | 68 | 68 | 0.811 |
| 14 | 182 | 2.170 | 30 | 30 | 0.358 | 69 | 69 | 0.823 |
| 15 | 150 | 1.788 | 32 | 64 | 0.763 | | | |

Notes: This table reports full sample distribution by the number of the portfolio composition. We report the number of observations. Panels span the period 2003–2017.

The dependent variable in this study was the Chinese local SOEs' CBM & A dummy and their proportion. The proportion was measured by Equation (1). This study set up the CBM & A dummy, being considered as "1" if in any year from 2003 to 2017 a particular enterprise had a state-owned merger and acquisition; otherwise, it is considered as "0". In the robustness test, we used Chinese enterprises' overseas revenues to compare the result with CBM & A. In addition, we changed the dependent variable to the overseas revenue to do the additional test

$$GI_{INDEXi.t,g} = \frac{VR_{i,t,g} * SIZE_{i,t,g}}{\sum_{i=1}^{n} VR_{i,t,g} * SIZE_{i,t,g}} \tag{1}$$

where $VR_{i,t,g}$ expresses, in the investment portfolio, $g$, of $t$ years, the ultimate control ratio of the controller to a company, $i$. $SIZE_{i,t,g}$ expresses the enterprise size of company $i$ in investment portfolio $g$ in $t$ years. $VR_{i,t,g} * SIZE_{i,t,g}$ expresses that the person with ultimate control can control the size of the company's resources. $GI\_INDEX_{i,t,g}$ expresses the proportion of company $i$ in portfolio $g$ in $t$ years. Boateng et al. [46] describe that the value of $GI\_INDEX_{i,t,g}$ reflects the importance of each

company in the portfolio of its ultimate controller. The larger the change value is, the more resources the ultimate controller can actually allocate through intervention in the portfolio of the company, so that the enterprise is more likely to be interfered with by the government. In the specific calculation, we used the data of total assets to represent the size of the enterprise ($TOTAL\ ASSET_{i,t,g}$) to calculate the degree of government intervention ($GI\_TA_{i,t,g}$).

$$GI\_TA_{i,t,g} = \frac{VR_{i,t,g} * TOTAL\ ASSET_{i,t,g}}{\sum_{i=1}^{n} VR_{i,t,g} * TOTAL\ ASSET_{i,t,g}} \tag{2}$$

Government internationalization subsidies are the data of the government that encourage foreign subsidies. These data were taken from the 2006–2017 CSMAR database. Due to the availability of data, we manually completed the data from 2003 to 2006 for the relevant listed companies' annual reports. This variable is widely mentioned in the scholars' future studies part [46].

Government intervention is the data of the shareholding ratio of the first state shareholder of enterprises that have been local SOEs. The data of this variable is from the 2003–2017 CSMAR database. This variable is widely mentioned by scholars' studies as being an intervening variable or independent variable [22,38].

We used the marketization index from 2003–2015 compiled by Fan et al. [8] to measure the strength of market influence. The above variable was used to measure the market institutional environment of the province where the government of the controlling region is located. We used interpolation to supplement the data from 2016–2017. The marketization index of Fan et al. [8] is a relatively comprehensive index. There are 26 indicators in total from five factors in China, including the role of the market, etc. The higher the score, the more market oriented the region. Fan's market index is already widely used by scholars (e.g., Berchicci [47]).

Referring to the practice of the literature, this paper sets the control variables as follows: First, we used the firm size as the control variable. The firm size was measured by the total number of employees in local SOEs [16,18]. Second, the firm age in this paper was measured by the number of years since the local SOEs were founded [41]. Thirdly, we used a ratio of total debts to total assets to express leverage [11]. Fourth, ROA was calculated by the total return divided to the total assets [31]. Fifth, human resources were operationalized using the natural logarithm of each local SOE's training expenditure for employees [45]. Sixth, we included Tobin's Q, which was calculated by the ratio of a firm's market value to its replacement cost of capital [14]. It was used to express the relationship between the value an enterprise creates and the cost of the assets it invests. Seventh, the R&D expenditure was the R&D expenditures for local SOEs and then the logarithm was taken [17]. The R&D expenditure expressed an enterprise's innovation ability and efficiency. Eighth, VR2 is the square of VR (VR2 represents the square of ownership concentration). It was used to estimate the non-linear influence of ownership concentration on the enterprise value [30]. Ninth, we included investing as a control variable. It was calculated by "payment for the purchase of fixed assets, intangible assets, and other long-term assets" minus "payment for the disposal of fixed assets, intangible assets, and other long-term assets" then divided by "total assets at the end of the period" [48]. It expresses the investment efficiency. The tenth control variable was board independence [49]. It was calculated by the number of independent directors divided by the total number of directors. It means the power of the board supervision. Eleventh, management shareholding was included to control the management level impact to CBM & A, which was calculated by "tradable shares held by management" added to "restricted sale shares held by management" and divided by "total share capital", and then the natural log was taken [28]. Twelfth, market competition was realized using the Herfindahl-Hirschman Index (HHI). HHI in this paper was calculated by the sum of the squared percentages of the total revenue or total assets of each market competitor in an industry. The data we used in this paper were from Shanghai and Shenzhen A-share data from 2003 to 2017 (excluding listed companies that were delisted by the end of 2017). According to the 2012 industry standard of the China Securities Regulatory Commission classification of the local SOEs' industry, the direct calculation is based on a

three-level classification, calculated on the basis of the top five enterprises in the industry in terms of operating income or total assets (which can be modified specifically). Finally, an executive dummy was defined when the chairman and general manager were the same people—the value was "1" and otherwise "0" [1]. The specific names, definitions, and symbols of the variables are shown in Table 3.

**Table 3.** Variable definitions.

| Variable | Definition | Expected | Symbol |
|---|---|---|---|
| CBM & A dummy/Overseas revenue | Local SOEs' CBM & A | | CBMA |
| Government internationalization subsidies | Government subsidies to local SOEs when they take CBM & A, the statistic for which was manually collected from the CSMAR database | + | GS_INTER |
| Government intervention | The shareholding ratio of the largest state shareholder | +/− | GI |
| Region-specific institutional development | Region-specific home institutional development (Fan Gang market index) | + | IC |
| Firm size | Number of employees | + | SIZE |
| Firm age | Number of years since establishment | + | AGE |
| Leverage | Ratio of total debts to total assets | + | LEV |
| ROA | Rate of assets | + | ROA |
| Human resources | Training expenditure per employee | + | HR |
| Tobin's Q | The ratio of a firm's market value to its replacement cost of capital | + | TOBINQ |
| R&D expenditures | Value of R&D expenditures | + | R&D |
| VR$^2$ | Control of the non-linear influence of ownership concentration on enterprise value | +/− | VR2 |
| Invest | (Cash paid for the purchase and construction of fixed assets, intangible assets, and other long-term assets—net cash received from the disposal of fixed assets, intangible assets, and other long-term assets)/total assets | + | INVEST |
| Board independence | Number of independent directors/total number of directors | +/- | BI |
| Management shareholding | (Tradable shares held by management + restricted sale shares held by management)/total share capital | + | MS |
| Market competition | The sum of the squared percentages of total revenue or total assets of each market competitor in an industry. Compiled using Shanghai and Shenzhen A-share data from 2003 to 2017 (excluding listed companies that have been delisted by the end of 2017) and excluding the financial and insurance industries. Industry description: according to the 2012 industry standard of the China Securities Regulatory Commission, according to the latest industry classification of the company, the direct calculation is based on a three-level classification. Calculated on the basis of the top five enterprises in the industry in terms of operating income or total assets (it can be modified specifically). | + | HHI |
| Executives dummy | Dummy is equal to "1" if the chairman and general manager are the same person and equal to "0" if the chairman and general manager are a different person. | +/− | EXE |

Table 4 shows that the average government intervention ratio of local SOEs was 40.27%. This is quite high but is not surprising because all firms in our sample were local state-owned enterprises. The average value of region-specific institutional development was 7.104. The average value of region-specific institutional development was higher than the average value of the market. Because these firms operate in traditional industry, and therefore have decaying life cycles, they must rely on the development of new products to remain competitive. This is because these firms are characterized by high levels of nationalization, and they can rely on developing new products to stay competitive. The average leverage of local SOEs was 50.694. This is relatively small compared to the population mean because SOEs usually have more conservative operating rules than other sorts of firms. The average Tobin's Q of local SOEs was 1.624. It is higher than the average value of 0.6 because SOEs usually have fewer replacement costs compared with other types of firms. The average executives dummy was 0.807, which was well higher than the average of 0.5. This is due to the chairmen and general managers

of nationalized companies often being appointed by local government officials and institutions and lacking independence. The reason why the number of observations in Table 4 was not always the same was that for all independent variables and control variables in this research the extremum was removed and first-order lag by Stata software was used. The reason why the number of Tobin's Q was small was that we synthesized the data in the CSMAR and Wind databases and removed the inconsistent values between the two databases.

**Table 4.** Descriptive statistics.

| Variable Name | Observation | Mean | Standard Deviation | Minimum | Maximum. |
|---|---|---|---|---|---|
| CBM & A dummy | 13422 | 0.201 | 0.409 | 0.000 | 1.000 |
| Government internationalization subsidies | 12527 | 13.230 | 1.849 | 6.687 | 16.141 |
| Government intervention | 12527 | 40.268 | 15.677 | 11.500 | 77.020 |
| Region-specific institutional development | 12527 | 7.104 | 1.825 | 2.870 | 10.920 |
| Firm size | 12527 | 4.351 | 7.088 | 0.000 | 43.654 |
| Firm age | 12527 | 14.248 | 6.205 | −0.373 | 31.663 |
| Leverage | 12527 | 50.694 | 25.806 | 0.000 | 144.869 |
| ROA | 12527 | 5.045 | 7.453 | −22.387 | 31.11 |
| Human resources | 12527 | 14.951 | 1.548 | 4.737 | 17.817 |
| Tobin's Q | 10682 | 1.624 | 1.694 | 0.151 | 10.945 |
| R&D expenditure | 12527 | 17.525 | 1.912 | 6.835 | 21.637 |
| $VR^2$ | 12527 | 0.1867 | 0.135 | 0.013 | 0.593 |
| Invest | 12527 | 0.036 | 0.051 | −0.649 | 0.593 |
| Board independence | 12527 | 0.358 | 0.055 | 0.182 | 0.571 |
| Management shareholding | 12527 | 10.933 | 2.192 | 1.099 | 15.380 |
| Market competition | 12527 | 0.001 | 0.002 | −0.003 | 0.016 |
| Executives dummy | 12527 | 0.807 | 0.395 | 0.000 | 1.000 |

Notes: Panels span the period 2003–2017.

## 4. Empirical Results

In this section, we further use the empirical data to test the impact factors on Chinese local CBM & As. To test the above four hypotheses, this research used dynamic panel and hierarchical moderated regression analyses with a probit specification [50]. In order to eliminate the effects of heteroscedasticity and endogeneity, the data of each variable were processed by the natural logarithm in this paper. Finally, in this paper's test, we fixed the year and industry effect in every part. Considering the lag effect of economic variables on M&A, this paper adopted first-order lag for all explanatory variables and control variables. The variance expansion factor (VIF) between model variables was also monitored. The experimental results show that the VIFs value interval was (1.00, 1.38) The VIF value was much lower than the acceptable 10, indicating that there was no obvious multicollinearity among the model variables. In addition, the correlation coefficient shown in Table 5 was less than 0.5, which means there was no obvious collinearity between variables.

$$CBM\&A = \beta_0 + \beta_1 GS_{it} + \beta_2 Control_{it} + \varepsilon_{it} \tag{3}$$

$$CBM\&A = \beta_0 + \beta_1 GS_{it} + \beta_2 GI_{it} + \beta_3 GS_{it} * GI_{it} + \beta_4 Control_{it} + \varepsilon_{it} \tag{4}$$

$$CBMA = \beta_0 + \beta_1 GI_{it} + \beta_2 Control_{it} + \varepsilon_{it} \tag{5}$$

$$CBMA = \beta_0 + \beta_1 GI_{it} + \beta_2 IC_{it} + \beta_3 GI_{it} * IC_{it} + \beta_4 Control_{it} + \varepsilon_{it} \tag{6}$$

**Table 5.** Correlation matrix.

| Variable Name | 1 | 2 | 3 | 4 | 5 | 6 | 7 | 8 | 9 |
|---|---|---|---|---|---|---|---|---|---|
| 1 CBM & A dummy | 1.000 | | | | | | | | |
| 2 Government internationalization subsidies | 0.277 | 1.000 | | | | | | | |
| 3 Government intervention | −0.366 | −0.004 | 1.000 | | | | | | |
| 4 Region-specific institutional development | 0.435 | −0.098 | −0.073 | 1.000 | | | | | |
| 5 Firm size | 0.245 | 0.376 | 0.029 | −0.041 | 1.000 | | | | |
| 6 Firm age | 0.004 | −0.124 | −0.165 | 0.214 | 0.136 | 1.000 | | | |
| 7 Leverage | 0.014 | 0.006 | −0.299 | −0.279 | 0.167 | 0.393 | 1.000 | | |
| 8 ROA | 0.070 | 0.245 | 0.539 | 0.133 | −0.084 | −0.185 | −0.517 | 1.000 | |
| 9 Human resources | −0.226 | −0.091 | −0.299 | −0.503 | −0.325 | −0.028 | 0.067 | −0.134 | 1.000 |
| 10 Tobin's Q | −0.214 | −0.108 | 0.508 | 0.139 | −0.371 | −0.241 | −0.485 | 0.767 | −0.126 |
| 11 R&D expenditure | 0.294 | 0.440 | 0.053 | 0.394 | 0.612 | 0.167 | −0.386 | 0.246 | −0.348 |
| 12 VR$^2$ | −0.303 | 0.099 | 0.487 | −0.055 | 0.146 | −0.165 | −0.297 | 0.576 | −0.322 |
| 13 Invest | −0.179 | −0.148 | −0.076 | −0.011 | −0.068 | 0.603 | 0.244 | −0.054 | 0.311 |
| 14 Board independence | −0.155 | 0.331 | 0.121 | −0.351 | 0.527 | −0.011 | 0.356 | −0.126 | 0.063 |
| 15 Management shareholding | 0.276 | 0.274 | 0.057 | 0.311 | 0.088 | −0.342 | −0.242 | 0.593 | 0.007 |
| 16 Market competition | −0.146 | −0.112 | 0.327 | 0.068 | −0.268 | −0.404 | −0.499 | 0.559 | 0.134 |
| 17 Executives dummy | −0.279 | −0.130 | −0.024 | −0.140 | −0.081 | −0.220 | −0.182 | 0.023 | −0.205 |
| **Variable Name** | **10** | **11** | **12** | **13** | **14** | **15** | **16** | **17** | |
| 10 Tobin's Q | 1.000 | | | | | | | | |
| 11 R&D expenditure | 0.116 | 1.000 | | | | | | | |
| 12 VR$^2$ | 0.487 | 0.126 | 1.000 | | | | | | |
| 13 Invest | −0.137 | 0.082 | −0.075 | 1.000 | | | | | |
| 14 Board independence | −0.310 | 0.026 | 0.174 | −0.057 | 1.000 | | | | |
| 15 Management shareholding | 0.396 | 0.425 | 0.123 | −0.020 | −0.022 | 1.000 | | | |
| 16 Market competition | 0.570 | 0.027 | 0.344 | −0.083 | −0.232 | 0.486 | 1.000 | | |
| 17 Executives dummy | 0.244 | 0.070 | −0.030 | 0.090 | −0.360 | 0.105 | 0.142 | 1.000 | |

Note: All the correlation coefficients are statistically different from zero at the 1% significance level. Panels span the period 2003–2017.

Model 1–Model 4 in Table 6 show that the results for hypotheses such as Equations (3) and (4) (H1, H2a, and H2b). Model 5–Model 8 in Table 6 show the results for hypotheses such as Equations (5) and (6) (H2a, H2b, and H3). The effect of government internationalization subsidies and government intervention having a positive and significant impact to local SOEs' CBM & A separately, no matter whether government interaction existed, is shown in Table 6. When the government intervention and region-specific institutional development interaction effect existed separately, the interaction term had a negative coefficient. In addition, when there were no interaction variables, the firm age, firm size, and R&D expenditure were a significant and positive impact on local SOEs CBM & A. The interaction variables of firm size, R&D expenditure, invest, management shareholding, and market competition, were a significant and positive impact on local SOEs CBM & A. Under the same number of control variables, the R-square in the interaction existing model was higher than in the interaction inexistent model. The R-square gradually increased as the number of control variables increased. All of this is consistent with common sense. This further confirms that the model construction in this paper is scientific and reasonable.

**Table 6.** Hierarchical moderated regression of cross-border mergers and acquisitions (CBM & A) and government internationalization subsidies: panel probit estimation.

| | CBM & A Dummy | | | | | | | |
|---|---|---|---|---|---|---|---|---|
| | 1 | 2 | 3 | 4 | 5 | 6 | 7 | 8 |
| **Independent variable** | | | | | | | | |
| Government Internationalization subsidies (H1) | 0.6702 | 11.4770 ** | 39.4805 *** | 22.1475 ** | | | | |
| | (0.30) | (2.37) | (3.10) | (2.19) | | | | |
| Government intervention (H2a) | | 3.2554 * | | 7.5784 * | 0.0137 | 0.5414 *** | 1.2826 *** | 0.2860 *** |
| | | (1.95) | | (1.88) | (0.42) | (3.17) | (3.75) | (2.93) |
| Region-specific institutional development | | | | | | 2.4663 *** | | 1.3145 ** |
| | | | | | | (2.88) | | (2.30) |
| **Interactions variable** | | | | | | | | |
| Government internationalization subsidies * Government intervention (H2b) | | −0.2539 ** | | −0.5882 ** | | | | |
| | | (−2.03) | | (−1.91) | | | | |
| Government intervention * Region-specific institutional development (H3) | | | | | | −0.0722 ** | | −0.0325 ** |
| | | | | | | (−3.15) | | (−2.40) |
| **Control variable** | | | | | | | | |
| Firm size | | | −0.0004 * | −0.0001 | | | −0.0001 | 0.0001 *** |
| | | | (−1.89) | (−0.50) | | | (−0.13) | (3.02) |
| Firm age | | | 0.8440 ** | 0.3548 *** | | | −0.0001 | 0.1699 ** |
| | | | (2.79) | (4.60) | | | (−0.13) | (2.26) |
| Leverage | | | −0.1184 | −0.8722 | | | −0.0444 | −0.0084 |
| | | | (−1.41) | (−0.13) | | | (−0.85) | (−0.58) |
| ROA | | | 1.6711 | −0.2403 ** | | | −0.1966 | 0.1162 ** |
| | | | (0.53) | (−2.64) | | | (−1.43) | (2.44) |
| Human resources | | | 2.8733 * | 0.8739 | | | 1.6744 | 0.3270 |
| | | | (2.19) | (0.87) | | | (1.22) | (1.38) |
| Tobin's Q | | | −4.5011 *** | −1.0823 * | | | −2.3898 *** | −1.9233 ** |
| | | | (−2.68) | (−1.98) | | | (−2.64) | (−2.73) |
| R&D expenditure | | | 4.5011 ** | 1.2359 *** | | | 1.8839 *** | 0.0329 ** |
| | | | (3.13) | (5.02) | | | (3.39) | (2.68) |
| VR2 | | | 12.6633 | 1.8738 * | | | 0.2993 * | 0.5289 * |

**Table 6.** *Cont.*

| | | | | CBM & A Dummy | | | | |
|---|---|---|---|---|---|---|---|---|
| | 1 | 2 | 3 | 4 | 5 | 6 | 7 | 8 |
| | | | (1.40) | (2.04) | | | (2.13) | (1.97) |
| Invest | | | 12.6633 | 3.9827 ** | | | 4.9033 * | 7.2957 |
| | | | (1.40) | (2.83) | | | (1.83) | (0.83) |
| Board independence | | | −28.2834 | −5.2207 * | | | −2.9883 | −6.3934 |
| | | | (−1.30) | (−1.67) | | | (1.24) | (−0.73) |
| Management shareholding | | | 2.0438 *** | 0.3290 | | | 0.4726* | −0.0841 |
| | | | (3.43) | (2.27) | | | (1.72) | (−0.47) |
| Market competition | | | −2.0991* | −3.9096 | | | −7.0126 | 19.8794 |
| | | | (−2.02) | (−0.32) | | | (−0.92) | (0.07) |
| Executives dummy | | | −8.7412 | −4.9902 * | | | −3.8056 * | −2.3017*** |
| | | | (−2.48) | (−2.15) | | | (−1.71) | (−2.63) |
| Year-fixed effects | Y | Y | Y | Y | Y | Y | Y | Y |
| Industry-fixed effects | Y | Y | Y | Y | Y | Y | Y | Y |
| Observations | 12,527 | 12,527 | 12,527 | 12,527 | 12,527 | 12,527 | 12,527 | 12,527 |
| Log likelihood | −11.103 | −9.712 | −6.324 | −6.833 | −587.578 | −50.715 | −49.368 | −72.338 |
| $\chi^2$-statistic | 0.000 | 0.000 | 0.0000 | 0.0000 | 0.000 | 0.0000 | 0.0000 | 0.000 |
| Pseudo-R2 | 0.665 | 0.747 | 0.767 | 0.781 | 0.226 | 0.336 | 0.774 | 0.894 |

Note: The table reports estimated coefficients and standard errors (in brackets). The dependent variable is the cross-border M&A dummy. ***, **, and * denote the levels of statistical significance at 1%, 5%, and 10%, respectively. Panels span the period 2003–2017.

Table 7 describes the two-step GMM IV used to do the robustness test (Two-step GMM IV is a method commonly used to process panel data and is often used for robustness test in empirical studies.). We used two efficient instrumental variables estimators via the two-step GMM to do the robustness test. This method considers the influence of possible econometric problems, such as endogeneity, heterogeneity, and autocorrelation, on the results of the model estimation (Caldera, 2010). In this study we chose the industry average firm age and industry average firm size as instruments (rather than controls). Therefore, in order to test the correlation of instrumental variables and the applicability under exogenous conditions, two tests were carried out in this study by referring to the methods in the literature of Lee and Wu (2017). First, the Durbin-Wu-Hausman (DWH) test was performed, and the conclusion was that the null hypothesis—that the tool was not needed at the 1% significance level—was rejected [51]. Secondly, the Sargan test in the DWH test was carried out, and the results showed that the null hypothesis unrelated to instrumental variables and error terms was not excluded [52]. Finally, this study used the instrument redundancy (IR) test to confirm that the above two instruments were appropriate for the model, which confirmed that our choice was correct. The results of the models in Table 7 show that all the key results from the two-step GMM instrumental variables estimator, government intervention, and the mediation effect of the market institution context remained qualitatively unchanged. The results above were a good match with those in Table 6. However, the results in Model 1 and Model 2 conflicted with the above test. The results above can be explained by the fact that the industry average firm age and size that impact the effect from government internationalization subsidies and government intervention to local SOEs overseas revenue changed negatively.

**Table 7.** Robustness analysis: two-step GMM instrumental variables (IV) estimation.

| | Local SOEs Overseas Revenue | | | |
|---|---|---|---|---|
| | **1** | **2** | **3** | **4** |
| Independent variable | | | | |
| Government internationalization subsidies (H1) | −6.9733 *** | −1.2010 *** | | |
| | (−2.74) | (−3.65) | | |
| Government intervention (H2a) | −2.4641 *** | −0.5725 *** | 1.7801 *** | 0.9506 |
| | (−2.87) | (−4.85) | (4.42) | (3.03) |
| Region-specific institutional development | | | 10.1757 *** | 4.7217 *** |
| | | | (4.55) | (3.25) |
| Interactions | | | | |
| Government internationalization subsidies * Government intervention (H2b) | 0.1839 *** | 0.0439 *** | | |
| | (2.86) | (4.65) | | |
| Government intervention * Region-specific institutional development (H3) | | | −0.2483 *** | −0.1433 *** |
| | | | (−4.37) | (−3.23) |
| Control variable | | | | |
| Firm size | | 0.0001 | | 0.0002 * |
| | | (1.24) | | (1.84) |
| Firm age | | 0.1381 | | 1.0553 |
| | | (1.56) | | (1.29) |
| Leverage | | 0.0380 *** | | 0.0196 |
| | | (4.43) | | (1.48) |
| ROA | | −0.0566 *** | | 0.0791 |
| | | (−3.32) | | (1.48) |
| Human resources | | 0.4377 | | 0.2908 |
| | | (1.05) | | (1.33) |
| Tobin's Q | | −0.5634 * | | −0.1869 |
| | | (−2.04) | | (-1.33) |
| R&D expenditure | | 0.4582 * | | 0.8382 * |
| | | (2.04) | | (1.96) |

**Table 7.** *Cont.*

| | Local SOEs Overseas Revenue | | | |
|---|---|---|---|---|
| | **1** | **2** | **3** | **4** |
| VR$^2$ | | −1.1392 ** | | −0.4983 |
| | | (−2.31) | | (−0.93) |
| Invest | | 7.9087 *** | | 20.8017 *** |
| | | (4.89) | | (3.81) |
| Board independence | | 2.6567 *** | | 4.9356 |
| | | (5.98) | | (0.92) |
| Management shareholding | | 0.1999 ** | | 0.0420 |
| | | (2.37) | | (0.32) |
| Market competition | | 2.0590 ** | | 78.6575 |
| | | (2.56) | | (0.66) |
| Executives dummy | | 2.1365 ** | | −0.6735 |
| | | (2.38) | | (−1.05) |
| Observations | 12527 | 12527 | 12527 | 12527 |
| F-statistic | 6.050 ** | 115.550 *** | 13.050 *** | 322.000 *** |
| Endogeneity test | 5.073 *** | 5.040 *** | 10.897 *** | 4.616 *** |
| Instrument redundancy test | 4.702 *** | 10.627 *** | 14.002 *** | 5.609 *** |

Note: Cluster year and industry. Firm age and firm size of industry average as instruments used in GMM test. The table reports estimated coefficients and standard errors (in brackets). The dependent variable is the local SOE's overseas revenue. ***, **, and * denote the levels of statistical significance at 1%, 5%, and 10%, respectively. Panels span the period 2003–2017.

As mentioned above, there was an obvious "self-selection effect" when SOEs made outbound investments or mergers and acquisitions. The existence of this "self-selection effect" made it necessary to select appropriate control group enterprises for verification when studying the influence of external factors on state-owned enterprises' CBM & A. A sample matching algorithm was the most appropriate construction method of the control group. This kind of algorithm, through the relevant procedures for extracting control sample structure as big as the experimental group in the control group, the control group in every way, and the experimental group, can not only reduce the confounding variables, but the results can also reduce the data selection bias, which is used to test the process better. Therefore, following Yi et al. [50], this study used PSM to select control group enterprises for the experimental group, making the regression results more reliable.

With reference to the summary of Caliendo and Kopeinig [53], we firstly used a logit model to estimate the propensity score of an enterprise's government subsidies. In this study, government subsidies were set to 1, otherwise 0 was used. The selected variables included leverage, human resources, Tobin's Q, R&D expenditure, and the return on assets. Government intervention was set to 1 when bigger than the median, otherwise is was set to 0. The selected variables included leverage, human resources, industry, board independence, management shareholding, and executive. The calculation method can be expressed as $Logit(GS = 1)\varnothing H_{i(t-1)}$. Secondly, according to the calculation of the propensity score, the nearest neighbor method was used to match the experimental group with the control group. According to the above, after the calculation of the propensity score, the nearest distance method was used to match the enterprises with and without government subsidies (Model 1) and without government intervention by a 1 to 1 ratio (Model 2), while the matched control group was selected. The specific distribution is shown in Tables 8 and 9.

**Table 8.** Description of government subsidies and government intervention in the control group.

| Year | 2003 | 2004 | 2005 | 2006 | 2007 | 2008 | 2009 | 2010 |
|---|---|---|---|---|---|---|---|---|
| Government Subsidies Events | 0 | 0 | 0 | 0 | 0 | 1 | 4 | 5 |
| Proportion | 0.000 | 0.000 | 0.000 | 0.000 | 0.000 | 0.006 | 0.024 | 0.029 |
| Government Intervention Events | 0 | 39 | 100 | 105 | 120 | 129 | 116 | 116 |
| Proportion | 0.000 | 0.030 | 0.076 | 0.080 | 0.092 | 0.099 | 0.089 | 0.089 |
| year | 2011 | 2012 | 2013 | 2014 | 2015 | 2016 | 2017 | Total |
| Government Subsidies Events | 6 | 6 | 37 | 39 | 34 | 38 | 0 | 170 |
| Proportion | 0.035 | 0.035 | 0.218 | 0.229 | 0.200 | 0.224 | 0.000 | 1.000 |
| Government Intervention Events | 124 | 128 | 136 | 94 | 41 | 34 | 26 | 1308 |
| Proportion | 0.095 | 0.098 | 0.104 | 0.072 | 0.031 | 0.026 | 0.020 | 1.000 |

Note: Panels span the period 2003–2017.

**Table 9.** Propensity score matching (PSM) result of government subsidies and government intervention.

| Variable Name | 1 | 2 |
|---|---|---|
| Leverage | 0.025 * | 0.0039 * |
| | (1.85) | (2.24) |
| ROA | 0.0193 | 0.0009 |
| | (0.57) | (0.09) |
| Human resources | −0.1726 ** | |
| | (−2.86) | |
| Tobin's Q | 0.0286* | |
| | (1.95) | |
| R&D expenditure | 0.4789 *** | |
| | (3.72) | |
| Board independence | | 0.1415 *** |
| | | (4.74) |
| Management shareholding | | 2.3720 ** |
| | | (3.22) |
| Executives dummy | | −12.3074 *** |
| | | (−2.71) |
| Industry | | −0.0231 *** |
| | | (−2.60) |
| Year-fixed effect | Y | Y |
| Industry-fixed effect | Y | Y |
| Observations | 12233 | 12365 |
| Log likelihood | −93.648 | −728.305 |
| $\chi^*$-statistic | 0.0000 | 0.000 |
| Pseudo-R* | 0.474 | 0.507 |
| ATT | 0.9673 *** | 0.8947 *** |
| ATE | 0.8479 *** | 0.8213 *** |

Note: The table reports estimated coefficients and standard errors (in brackets). ***, **, and * denote the levels of statistical significance at 1%, 5%, and 10%, respectively. Panels span the period 2003–2017. ATT means the average treatment effect on the treated of the PSM model. ATE means the average treatment effect on the population of PSM model.

In Table 9, we used the result of PSM in government internationalization subsidies (Model 1 and Model 2) and government intervention (Model 3 and Model 4) to check the regression result, respectively. Compared with the result in Table 6, the result in Table 10 had relatively poor significance. This is likely to be because the sample of the PSM result was too small to explain the CBM & A. The other explanation of this was that non-SOEs were less affected by government involvement, while the coefficient of the independent variable in Table 10 was the same as above.

**Table 10.** Result of PSM government subsidies and government intervention: panel probit estimation.

| | Cross-Border M&A Dummy | | | |
| --- | --- | --- | --- | --- |
| | **1** | **2** | **3** | **4** |
| Independent variable | | | | |
| PSM Government internationalization subsidies (H1) | 4.6852 | 3.2381 * | | |
| | (0.88) | (1.85) | | |
| PSM Government intervention (H2a) | 2.7042 * | 0.6720 ** | −0.1737 * | −0.0710 ** |
| | (1.87) | (2.43) | (1.83) | (−2.12) |
| Region=specific institutional development | | | 0.0328 ** | −0.2557 * |
| | | | (2.55) | (−1.86) |
| Interactions | | | | |
| PSM Government internationalization subsidies * PSM Government intervention (H2b) | −0.1847 ** | −0.2873 ** | | |
| | (−2.65) | (−2.44) | | |
| PSM Government intervention * Region-specific institutional development (H3) | | | 0.0023 * | 0.0104 ** |
| | | | (1.99) | (2.27) |
| Control variable | | | | |
| Firm size | | 0.3895 | | 0.0685 ** |
| | | (1.57) | | (2.34) |
| Firm age | | 0.0624 *** | | 0.0685 ** |
| | | (5.29) | | (2.34) |
| Leverage | | 0.0023 * | | 1.0979 * |
| | | (1.79) | | (1.95) |
| ROA | | 8.7954 * | | 5.8933 *** |
| | | (2.39) | | (6.94) |
| Human resources | | −0.7862 ** | | −0.1782 * |
| | | (−2.02) | | (−1.99) |
| Tobin's Q | | −1.3922 *** | | −0.2108** |
| | | (−4.23) | | (−2.16) |
| R&D expenditure | | 11.2307 ** | | 0.3795 *** |
| | | (2.78) | | (4.51) |
| VR* | | 2.3109 | | 1.1967 |
| | | (0.87) | | (1.58) |
| Invest | | −0.0098 * | | −4.8991 * |
| | | (−1.99) | | (−1.79) |
| Board independence | | −0.9817 | | −2.2826 |
| | | (1.02) | | (−0.85) |
| Management shareholding | | 3.9822 *** | | 1.0948 ** |
| | | (4.29) | | (2.83) |
| Market competition | | 8.4591 * | | 4.9331 ** |
| | | (1.80) | | (2.39) |
| Executives dummy | | 1.9033 * | | 1.7833 *** |
| | | (1.90) | | (5.29) |
| Year-fixed effect | Y | Y | Y | Y |
| Industry-fixed effect | Y | Y | Y | Y |
| Observations | 12233 | 12233 | 12365 | 12365 |
| Log likelihood | −4.091 | −3.128 | −677.747 | −89.418 |
| χ *-statistic | 0.000 | 0.000 | 0.000 | 0.000 |
| Pseudo-R * | 0.226 | 0.371 | 0.274 | 0.575 |

Note: The table reports estimated coefficients and standard errors (in brackets). The dependent variable is the cross-border M&A dummy. ***, **, and * denote the levels of statistical significance at 1%, 5%, and 10%, respectively. Panels span the period 2003–2017.

Then, we used the CBM & A proportion as the dependent variable to test the rationality of using the dummy variables above. In Model 1 and Model 2 in Table 11, we tested the impact of government internationalization subsidies on local SOEs' CBM & A proportions, and in Model 3 and Model 4, we tested the impact from government intervention on local SOEs' CBM & A proportions. The result of the regression supports the result in the main regression. Model 2 and Model 4 include the control variables. What is noticeable is that the firm age and investment had a positive coefficient when the dependent variable was local SOEs' CBM & A proportions.

**Table 11.** Robustness analysis of state-owned enterprises (SOEs) CBM & A: panel OLS (Ordinary Least Squares) estimation.

| | Cross-Border Proportion | | | |
| --- | --- | --- | --- | --- |
| | **1** | **2** | **3** | **4** |
| Independent variable | | | | |
| Government internationalization subsidies (H1) | 0.4418 *** | 0.6565 ** | | |
| | (8.83) | (2.52) | | |
| Government intervention (H2a) | 0.0301 * | 0.1894 * | 0.0092 * | 0.0858 ** |
| | (1.83) | (1.82) | (1.93) | (2.19) |
| Region-specific institutional development | | | 0.1309 ** | 0.1003 * |
| | | | (2.80) | (1.89) |
| Interactions | | | | |
| Government internationalization subsidies * Government intervention (H2b) | −0.0019 * | −0.0168 ** | | |
| | (−1.96) | (−2.17) | | |
| Government intervention * Region-specific institutional development (H3) | | | −0.0007 ** | −0.0093 * |
| | | | (2.47) | (1.89) |
| Control variable | | | | |
| Firm size | | 0.0001 | | 0.0001 |
| | | (0.90) | | (0.94) |
| Firm age | | 0.0764 * | | 0.0198 |
| | | (1.73) | | (0.67) |
| Leverage | | −0.0091 | | 0.0335 *** |
| | | (−0.59) | | (3.18) |
| ROA | | 0.0999 ** | | 0.0177 |
| | | (2.27) | | (0.44) |
| Human resources | | 0.0037 | | 0.3291 |
| | | (1.32) | | (0.87) |
| Tobin's Q | | −0.8032 *** | | −0.3401 |
| | | (−3.32) | | (−1.26) |
| R&D expenditure | | 0.2577 | | 0.4540 *** |
| | | (1.41) | | (3.94) |
| VR * | | 2.9033 * | | 0.9223 ** |
| | | (2.19) | | (2.58) |
| Invest | | 6.2193 ** | | 0.0023 * |
| | | (2.08) | | (1.94) |
| Board independence | | −4.9880 ** | | −4.2993* |
| | | (−2.35) | | (−2.04) |
| Management shareholding | | −0.0093 | | −0.0025 |
| | | (−0.13) | | (−0.06) |
| Market competition | | −5.2722 ** | | −0.1718 |
| | | (−3.04) | | (−0.08) |
| Executives dummy | | 0.3526 | | 0.4892 * |
| | | (1.11) | | (1.90) |
| Year-fixed effect | Y | Y | Y | Y |
| Industry-fixed effect | Y | Y | Y | Y |
| Observations | 1027 | 1027 | 1027 | 1027 |
| F-statistic | 4.142 *** | 18.475 *** | 6.668 *** | 32.118 *** |
| Adjusted R * | 0.134 | 0.638 | 0.128 | 0.688 |

Note: Cluster year and industry. The table reports estimated coefficients and standard errors (in brackets). The dependent variable is the local SOE's overseas revenue. Results are corrected for heteroskedasticity. ***, **, and * denote the levels of statistical significance at 1%, 5%, and 10%, respectively. Panels span the period 2003–2017. OLS (Ordinary Least Squares) is a mathematical optimization technique. It finds the best functional match for the data by minimizing the sum of squares of errors.

## 5. Finding and Discussion

Based on the above empirical conclusions, it can be seen that government internationalization subsidies significantly and positively affect local SOEs' CBM & A separately. Additionally, government intervention significantly and positively affects local SOEs' CBM & A separately.

On the other hand, when government intervention exists, the impact from government internationalization subsidies to local SOEs CBM & A weakens, and even this effect can become

negative. This is due to the appropriate government fiscal subsidy, which is conducive to local state-owned enterprises making strong choices of overseas M & A [4,22,30]. However, when there are government subsidies and large local government stakes, this advantage is offset or it even hinders overseas mergers and acquisitions. In addition, when a market institution context exists, the impact from government intervention to local SOEs' CBM & A weakens, and even this effect can become negative. This is because appropriate government financial and personnel intervention is conducive to the development of SOEs' overseas business and mergers and acquisitions' activities [3,40]. Whereas, when local SOEs are located in a place with a high degree of marketization, the high market institution context dilutes the advantages of government intervention, so far as the marketization mechanism produces contradiction with government intervention, which is shown by executive delegation [33,45], the tax system [28], and firm management systems [54,55].

Another concerning point are the coefficients of the control variables. The vast majority of the coefficients of control variables match with what was expected in this study, with the exception of firm size, human resource, market competition, and invert. The coefficient of firm age is negative, because the government binding of SOEs is enhanced when they go through it for a long time, which impacts on their operation efficiency and executive judgment, etc. [10]). The coefficient of human resource is negative because the training of SOEs' employees lean toward formalism and bureaucracy, which allows an employee of an SOE to have resistance and a disgusted emotion toward the training. The coefficient of market competition is negative because much of the competitiveness of SOEs comes from local government resources and support. If the competition is strong, the binding and control power from local government authority is stronger. The above phenomenon weakens creativity and development power [21,56]. The coefficient of investment is negative—this paper presents this as "invest"—which means the ability and efficiency of local SOEs investment return. The negative coefficient of investment is due to the operational efficiency of local SOEs usually being lower than other types of firms due to cumbersome audit processes and unreasonable personnel appointment systems. Hence, the vaster the investment of local SOEs, the larger the firm's occupied cash flow. This will lead them to not being able to have enough cash flow to proceed with overseas M&A [57]. The efficient of board independence is negative. This is the other interesting finding in the main regression model. This is because the independent director in local SOEs usually has some interest or relationship with local government [14]. Therefore, they cannot truly supervise local government-controlled enterprises fairly.

In an additional test, we used different dependent variables, which included overseas revenue. Meanwhile, we also used other methods (two-step GMM IV) to verify the models' accuracy. What is noticeable is that we used the propensity score match (PSM) to match the control group of government internationalization subsidies and government intervention, then used those two new variables to test the robustness of the independent variables in the main regression. The result of the robustness test shows that the result in the main test is robust. Second, the control variables also significantly impact local SOEs' CBM & A.

## 6. Conclusions

We sought to examine whether and how government involvement influences local SOEs' CBM & A activities, and by what factors this influence is moderated. By treating the market institution context as an endogenous factor, government internationalization subsidies and government intervention are two integral components of government involvement. Our findings show that government internationalization subsidies and government intervention have a positive and significant impact on government involvement, separately. However, government intervention negatively and significantly moderates the effect of government internationalization subsidies on local SOEs' CBM & A. The market institution context negatively and significantly moderates the effect of government invention on local SOEs CBM & A. The theoretical and practical contributions of this study include: (1) This study fills the theoretical gap on the influence of different and various ways of government participation on the cross-border mergers and acquisitions of local SOEs. (2) This study provides the theoretical

contribution of the regulating effect of the institutional context between government involvement and local SOEs' cross-border mergers and acquisitions. (3) The conclusion of this study makes it more reasonable for the government to use different methods to participate in the cross-border M&A of local SOEs. This provides a practice contribution. (4) Another practice contribution is that we found out about the influence of many control variables on government intervention in transnational mergers and acquisitions of local SOEs. This provides a practical contribution to the government's differentiated intervention in the cross-border mergers and acquisitions of local SOEs with different characteristics.

Hence, we can see that government involvement has a positive impact on local SOEs' CBM & A. However, if too much government involvement adds up, the beneficial effects of government involvement weaken, and even become counterproductive. In addition, when local governments are involved in local SOEs, they should avoid their behavior conflicting with market rules or market events. To help and assist local SOEs to improve the CBM & A success rate, in terms of the regression result of control variables, the government should pay attention to the high-age local SOEs' CBM & A behavior. They should properly broaden the control of those local SOEs. The government should consider the practice of their employee training to avoid formalism. The government should also maintain normal market order and try to avoid using administrative directives to interfere with market rules to push local SOEs' continuous development through the pressure to survive. Local government should also speed up the updating of investment of local SOEs. It is explained here that the data set used in this paper includes the data of all listed companies in the Chinese market, so the conclusion of this study can be applied to any region of China.

Although the research design in this paper is solid, it also has some limitations. Here, some suggestions for the improvement of future research and the possibility of extending the research results are given. First, we measure SOEs' CBM & A based on whether firms have done CBM & A in this period. Our experiment cannot assess the degree of the impact of CBM & A. Therefore, we cannot determine the degree and strength of the effect, or if there was no effect, of government internationalization subsidies and government intervention on the local SOEs' CBM & A. Second, this paper only uses samples of local SOEs in China. This may lead to questions about the applicability of the research results of this paper—mainly about whether they can be applied to other countries. This is also because the government intervention that forms the main body of the research model in this paper are different according to the characteristics and regulatory degree of the region and industry. Therefore, data from other countries could be used to verify the model in the future. Third, an interesting point for future research is the tension over the importance of CBM & A for external institutions in different industries and regions. Fourth, we used industry average age and size as instrument variables. In fact, the local SOEs in specific industries and years were low in number, which means the instrument variables have, to some extent, correlation with every single firm. Hence, in future research, we could use the provincial GDP (Gross Domestic Product) or CPI (Consumer Price Index)index to avoid this problem. Finally, we selected all local SOEs as the sample. On the other hand, we could classify the sample firms by their shareholder local governments' administrative level (province, city, and county). This method could help us to analyze whether the different levels of government have different power to impact local SOEs' CBM & A.

**Author Contributions:** Conceptualization, Q.G. and F.B.; methodology, Q.G., C.J., and F.B.; software, Q.G.; validation, C.J., Q.G., and F.B.; formal analysis, F.B.; investigation, Q.G.; data curation, Q.G.; writing—original draft preparation, Q.G. and F.B.; writing—review and editing, Q.G. and F.B.; visualization, Q.G.; supervision, C.J.; project administration, F.B.; funding acquisition, C.J. All authors have read and agreed to the published version of the manuscript.

**Funding:** This research was funded by the Natural Science Foundation of Zhejiang Province (Grant No. LQ20G010002), Zhejiang Provincial Key Project of Philosophy and Social Sciences (Grant No. 20NDJC10Z), and the National Science Foundation of China (71571162).

**Acknowledgments:** The authors also gratefully acknowledge the helpful comments and suggestions of the reviewers and Geyao Li and Yihao Jiang who improved the presentation.

**Conflicts of Interest:** The authors declare no conflict of interest.

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
