# Peer review of "The Cross-Border Mergers and Acquisitions of Local State-Owned Enterprises: The Role of Home Country Government Involvement"

_sustainability, doi:10.3390/su12073020_

Round 1

Reviewer 1 Report

I thank the authors for their work and the quality of the study presented. Minor changes are only mentioned in areas where manuscript could be clearer and more complete.

Comments below:

It is not clear how the control variables in Figure 1 were defined. What is their relevance? Why these and not others?

It is also unclear how the dataset built by the authors was obtained. What is the source database? What information is available in this database?

It would be important to highlight the reasons why the number of observations in Table 4 is not always the same.

Table 7 should be placed in page 14.

It would be important to better organize the “findings and discussion” section. Dividing the analysis between the various themes by different paragraphs would make it much easier to read.

The conclusions section should clarify the theoretical and practical contributions of this study.

The authors should clarify what is the potential of this study to be applied in other geographical areas of China.

Author Response

Dear Editors and Reviewers: Thank you for your letter and for the reviewers’ comments concerning our manuscript entitled “The Cross-Border Mergers and Acquisitions of Local State-owned Enterprises: The Role of Home Country Government Involvement” (ID: sustainability-739584). Those comments are all valuable and very helpful for revising and improving our paper, as well as the important guiding significance to our researches. We have studied comments carefully and have made correction which we hope to meet with approval. Revised portion are marked in revisions mode and red in the paper. The main corrections in the paper and the responds to the reviewers’ comments are as flowing: Responds to the reviewer’s comments: 1. Response to comment: It is not clear how the control variables in Figure 1 were defined. What is their relevance? Why these and not others? Response: Considering the Reviewer’s suggestion, we have added and modified content as “Figure 1 shows the brief model for this research. The government internalization subsides and government intervention in figure 1 represent the independent variable in this research. The Local State-owned Enterprises Cross-border Mergers and Acquisitions in figure 1 represents the dependent variable in this research. The institutional context (Region-specific home institutional development) in figure 1 represents the moderator variable in this research. The all variables in control variable part represent the control variable in this research. All the details of control variables and related literature are detailed in chapter 3.”We also modify and red the relevant paragraph in chapter 3“Referring to the practice of literature, this paper sets the control variables as follows. First, we used firm size to be control variable. The firm size is measured by the total number of employees in local SOEs (Xia & Chen, 2007; Chen, Li, Shapiro & Zhang, 2014). Second, firm age in this paper is measured by the number of years since the local SOEs was founded (Coluccia, Fontana and Solimene, 2018). Thirdly, we use ratio of total debts to total assets to express leverage (Zhao, Chen and Hao, 2018). Fourth, ROA is calculated by total return divided to total asset (Cui and Jiang, 2012). Fifth, human resources are operationalized using natural logarithm of each local SOEs training expenditure for employees (Li and Ding, 2013). Sixth, we include Tobin Q which calculated by the ratio of a firm's market value to its replacement cost of capital (Stan, Peng and Bruton, 2014). It is used to express the relationship between the value an enterprise creates and the cost of the assets it invests. Seventh, R&D expenditure is the R&D expenditures for local SOEs, and then take the logarithm (Garcia‐Canal and Guillén, 2008). The R&D expenditure express the enterprise innovation ability and efficiency. Eighth, VR2 is the square of VR. It is used to estimate the non-linear influence of ownership concentration on enterprise value (Lin, Cai and Li, 1997). Ninth, we include invest to be a control variable. It calculated by “payment for the purchase of fixed assets, intangible assets and other long-term assets” minus “payment for the disposal of fixed assets, intangible assets and other long-term assets”, then divided “total assets at the end of the period” (Brockman, Rui and Zou 2013). It expresses the investment efficiency. The tenth control variable is board independence (Volkery and Ribeiro, 2009). It calculated by the number of independent directors divided total number of directors. It means the power of board supervision. Eleventh, management shareholding is included to be control the management level impact to CBM&A, which calculated by “tradable shares held by management” add “restricted sale shares held by management” and divided to “total share capital,” and then take the natural log. (Busse and Hefeker, 2007). Twelfth, Market Competition is realized by Herfindahl-Hirschman Index (HHI) HHI in this paper calculated by the sum of the squared percentages of total revenue or total assets of each market competitor in an industry. The data we used in this paper is from Shanghai and Shenzhen A-share data from 2003 to 2017 (excluding listed companies that have been delisted by the end of 2017). According to the 2012 industry standard of China securities regulatory commission to classify the local SOEs’ industry, the direct calculation is based on the three-level classification. Calculated on the basis of the top five enterprises in the industry in terms of operating income or total assets (which can be modified specifically). Finally, executive dummy is defined when chairman and general manager are the same person, the value is “1”, otherwise “0” (Hao and Lu, 2018).” 2. Response to comment: It is also unclear how the dataset built by the authors was obtained. What is the source database? What information is available in this database? Response: Considering the Reviewer’s suggestion, we have added and modified content as “We constructed a dataset for local state-owned companies trading in the Shanghai and Shenzhen Stock Exchange that have been involved in outward FDI and exporting activities between 1998 and 2017 (see Table 1 for details). Information regarding Chinese local SOEs outward CBM&A activities and firm-specific information, such as firm age, size, number of employees, staff training expense, ownership structure, return on assets (ROA) and R&D expenditure, was obtained from CSMAR database. We extracted firm exporting and executives (name, age, and education background) data from Wind database. Information on different types of government subsidies, government intervention, ROA, asset-liability ratio, Tobin Q, export subsidies, financial subsidies, and other subsidies. What is worth paying attention to is Tobin Q selected in this paper is used by arithmetic mean. We also supplement it with another Chinese database reset and manually checked and supplement.” 3. Response to comment: It would be important to highlight the reasons why the number of observations in Table 4 is not always the same. Response: Considering the Reviewer’s suggestion, we have added and modified content as “The reason why the number of observations in Table 4 is not always the same is all independent variable and control variable in this research are remove the extremum and using first-order lag by Stata software. The reason why the number of Tobin Q few is that we synthesize the data in CSMAR and Wind database and remove the Inconsistent values between two databases.” 4. Response to comment: Table 7 should be placed in page 14. Response: Considering the Reviewer’s suggestion, we have moved Table 7 to the correct position. 5. Response to comment: It would be important to better organize the “findings and discussion” section. Dividing the analysis between the various themes by different paragraphs would make it much easier to read. Response: Considering the Reviewer’s suggestion, we have moved restructure the findings and discussion part for four paragraphs. 6. Response to comment: The conclusions section should clarify the theoretical and practical contributions of this study. Response: Considering the Reviewer’s suggestion, we have added and modified content as “The theoretical and practical contributions of this study including: (1) This study fills the theoretical gap on the influence of different and various ways of government participation on cross-border mergers and acquisitions of local SOEs. (2) This study provides the theoretical contribution of the regulating effect of institutional context between government involvement and local SOEs cross-border mergers and acquisitions. (3) The conclusion of this study makes it more reasonable for the government to use different methods to participate in the cross-border M&A of local SOEs. This provides the practice contribution. (4) Another practice contribution is that we find the influence of many control variables on government intervention on transnational mergers and acquisitions of local SOEs. This provides a practical contribution to the government's differentiated intervention in cross-border mergers and acquisitions of local SOEs with different characteristics.” 7. Response to comment: The authors should clarify what is the potential of this study to be applied in other geographical areas of China. Response: Considering the Reviewer’s suggestion, we have added and modified content as “It is explained here that the data set used in this paper includes the data of all listed companies in the Chinese market, so the conclusion of this study can be applied to any region of China.” Special thanks to you for your good comments. Finally, according to the suggestion of editors and experts, we have generally checked and adjusted the language of the article, corrected some grammatical errors, and improved the integrity of the article. We tried our best to improve the manuscript and made some changes in the manuscript. These changes will not influence the content and framework of the paper. We appreciate for Editors/Reviewers’ warm work earnestly, and hope that the correction will meet with approval. Once again, thank you very much for your comments and suggestions.

Reviewer 2 Report

An excellent and a very-interesting paper. The paper is quite interesting and significantly covers the particular settings and appropriate specifications of the Journal.

 Moreover, this paper is suitable for the contents and within research orientation of the Journal

The subjects of the paper are quite clear, well-structured and closely related with the aims of the Journal

The themes of the paper are relevant to this publication of the Journal.

The paper is well-constructed and also all themes are relevant and quite clearly stated. The paper demonstrates an adequate structure and uses the current literature in the field.

The paper reviews the current literature and the section of conclusions of this paper is quite clear, including the main – bullet points contributing towards the current theory.

Author Response

List of Responses:

Reviewers2: An excellent and a very-interesting paper. The paper is quite interesting and significantly covers the particular settings and appropriate specifications of the Journal.

Moreover, this paper is suitable for the contents and within research orientation of the Journal

The subjects of the paper are quite clear, well-structured and closely related with the aims of the Journal

The themes of the paper are relevant to this publication of the Journal.

The paper is well-constructed and also all themes are relevant and quite clearly stated. The paper demonstrates an adequate structure and uses the current literature in the field.

The paper reviews the current literature and the section of conclusions of this paper is quite clear, including the main – bullet points contributing towards the current theory.

Dear Editors and Reviewers: 

Thank you for your letter and for the reviewers’ comments concerning our manuscript entitled “The Cross-Border Mergers and Acquisitions of Local State-owned Enterprises: The Role of Home Country Government Involvement” (ID: sustainability-739584). Those comments are all valuable and very helpful for revising and improving our paper, as well as the important guiding significance to our researches. We have studied comments carefully and have made correction which we hope to meet with approval. Revised portion are marked in revisions mode and red in the paper. The main corrections in the paper and the responds to the reviewers’ comments are as flowing:

Responds to the reviewer’s comments:

Thank you for your recognition of our article. We have further improved the article according to the opinions of editors and experts. Special thanks to you for your good comments.

Finally, according to the suggestion of editors and experts, we have generally checked and adjusted the language of the article, corrected some grammatical errors, and improved the integrity of the article.

We tried our best to improve the manuscript and made some changes in the manuscript.  These changes will not influence the content and framework of the paper. We appreciate for Editors/Reviewers’ warm work earnestly, and hope that the correction will meet with approval.

Once again, thank you very much for your comments and suggestions.